# Factors associated with willingness to take Pre-Exposure Prophylaxis (PrEP) among high-risk adolescent boys and young men in Masese fishing community, Uganda

**Winnie Agwang**[1]*, **Joanita Nangendo**[2], **Sherifah Nabikande**[1], **Tom Okello**[2], **Joan Tusabe**[1], **Fred C. Semitala**[3,4], **Simon Kasasa**[1], **Joseph K. B. Matovu**[1,5]

1 School of Public Health, College of Health Sciences, Makerere University, Kampala, Uganda, 2 Clinical Epidemiology Unit, College of Health Sciences, Makerere University, Kampala, Uganda, 3 Department of Internal Medicine, Makerere University College of Health Sciences, Kampala, Uganda, 4 Makerere University Joint AIDS Program, Kampala, Uganda, 5 Faculty of Health Sciences, Busitema University, Mbale, Uganda

* winnieotuba@gmail.com

## Abstract

Pre-Exposure Prophylaxis (PrEP) is a known HIV prevention strategy for high-risk populations however, some high-risk communities have not yet fully embraced it. We sought to determine willingness to take PrEP and the associated factors among high-risk adolescent boys and young men (ABYM) in Masese fishing community, Jinja district, Eastern Uganda. We conducted a cross-sectional study, between October and November 2020, using a semi-structured questionnaire among ABYM aged 10–24 years in Masese fishing community, Eastern Uganda. We surveyed 479 participants, who had two or more sexual partners with inconsistent or no condom use. We carried out modified Poisson regression analysis to determine factors associated with willingness to take PrEP.Of 479 high-risk ABYM, 86.4% (n = 414) were willing to take PrEP. Confidence in PrEP safety (adj.PR = 1.56; 95%CI: 1.55, 2.24), availability of PrEP in areas easily accessible by ABYM (adj.PR = 1.40; 95%CI: 1.25, 1.57), and perceiving oneself as being at a very high risk of HIV infection (adj.PR = 1.11; 95%CI: 1.03, 1.20) were positively associated with willingness to take PrEP. On the other hand, being unmarried (adj.PR = 0.92; 95%CI: 0.87, 0.98) and earning more than USD 27 a month (adj.PR = 0.92; 95%CI: 0.87, 0.97) were negatively associated with willingness to take PrEP. There was high willingness to take PrEP among adolescent boys and young men in Masese fishing community. Confidence in PrEP safety, access to PrEP in their community and self-perception to be at high risk for HIV acquisition had a positive bearing on willingness to take PrEP while being unmarried and earning more than USD27 had a negative bearing on willingness to take PrEP. These findings suggest a need for target-specific interventions for unmarried men and those earning >USD27.

**Data Availability Statement:** All data are fully available without restriction with the study team.

**Funding:** Research reported in this publication was supported by the Fogarty international centre, National Institute of Alcohol Abuse and Alcoholism, National Institute of mental Health, of the National Institutes of Health under award number D43TW011304. The content is solely the responsibility of the authors and does not necessarily represent the official views of the National Institutes of Health. WA received the funding. The funders had no role in the study design, data collection and analysis, decision to publish or preparation of the manuscript.

**Competing interests:** The authors have declared that no competing interests exist.

## Introduction

Between 2010 and 2020, new HIV infections decreased by 31%, from 2.1million to 1.5million and HIV-related deaths by 47% [1, 2]. This achievement was due to implementation of combination HIV prevention interventions, including life-long antiretroviral treatment (ART), male circumcision, use of female and male condoms, behavior change communication and oral pre-exposure prophylaxis [3]. Pre-Exposure Prophylaxis (PrEP) is a biomedical HIV prevention intervention which has been demonstrated to be effective in preventing HIV acquisition [4]. PrEP is highly effective for preventing HIV when taken as prescribed by 75–99% [5]. The partners PrEP study in particular carried out in Uganda and Kenya showed an efficacy of 62% and 73% protection when TDF and TDF/FTC were used respectively [6]. Basing on extensive research, the World Health Organization (WHO) recommended PrEP for all populations at substantial risk of HIV infection [7]. However, PrEP effectiveness is dependent on adherence which is directly proportional to willingness to take PrEP [5, 8].

Willingness to take PrEP has been demonstrated by several studies to be between 46.5–90.4% among other high-risk groups [9–13]. Among fisher-folk community in peri-urban Kampala in Uganda, willingness to take PrEP was found to be 80.6% [14]. From previous research, barriers and facilitators to willingness to take PrEP among Men who have sex with men and transgender were drug use, having a full time job, regular screening for HIV, fear of risk compensation, cost of PrEP, side effects, mode of delivery (if it was injection or oral), medical mistrust, adherence issues, knowledge about PrEP, stigma, multiple sexual partnerships, casual sexual relationships and having condomless sex [9, 11, 12, 15–18], while among the opiate users, it was personal risk perception, cost of PrEP, side effects, mode of delivery (if it was injection or oral) [19] According to Muwonge et al., Ssuna et al., and Ddaaki et al., whose study populations included fisher folk willingness to take PrEP was associated with high risk self-perception, having had an HIV test in the last 6 months, tertiary education, Stigma, accessibility, side effects, schedules, were associated with willingness to take PrEP [14, 20, 21]. Among young people, the factors found to be associated with willingness to take PrEP were side effects, social stigma, health worker attitude, PrEP cost, accessibility and mode of administration [22]. These findings suggest that different factors apply to different populations given the different participant ages in the different studies, geographical locations, access to PrEP services among others and therefore may not be applicable to other populations despite the fact that they are all high-risk populations making implementation difficult. Fishing communities in particular are characterized by high mobility which forces men to stay away from their families for extended periods [23, 24] making them vulnerable to HIV. Men in the fishing communities also engage in sex with women in exchange for fish given that they have disposable income and can afford to pay [25–27] usually without using a condom [24, 28]. These increases men's risk of HIV infection [25, 29] which contributes to high prevalence of HIV in fishing communities with prevalence among men, as high as 18.7–37.0% [30–33], a prevalence much higher than that among men in the general population which is at 4.7% [34].

In a bid to reduce new HIV infections, PrEP distribution in Jinja district started in October 2019 though Walukuba Health Centre IV which serves Masese fishing community, started offering PrEP in November 2019. However, PrEP in fishing communities is not yet fully embraced despite having HIV prevalence as high as 37% [32] and ABYM are often times not the focus of research yet they could benefit from interventions such as oral PrEP together with their sexual partners. Therefore, understanding willingness to take PrEP among adolescent boys and young men in a fishing community will provide essential pre-intervention information that can guide the design and implementation of PrEP interventions for ABYM in other

fishing communities. The purpose of this study was to determine the willingness to take PrEP and associated factors among high-risk ABYM in Jinja district, Eastern Uganda.

### Theoretical basis of the study

The health belief model (HBM) in totality or in part is one of the models that can help understand individual and behavioural factors associated with willingness to take PrEP and ways to facilitate willingness to take PrEP. This model theorizes that a person's health behavior, in this case, willingness to take PrEP, is dependent on four areas of perception: (i) Perceived severity of the illness; (ii) their susceptibility to acquiring the illness; (iii) the benefits of taking a preventive action; and (iv) perceived barriers to taking the preventive action. Additionally, according to the HBM, cues to action influence behavior and move people to take on the preventive action [35]. Studies have found the HBM useful as a guiding framework for understanding the role of PrEP and HIV-related beliefs, concerns, and perceptions of risk [35].

## Materials and methods

### Study setting, population and sampling methods

We conducted a cross-sectional study in Masese Fishing community, Eastern Uganda between October and November 2020 among high-risk ABYM in the general population. Masese fishing community is located in Butembe suburb, East of Jinja Town with an estimated population of 34,000 people; of these, 55.5% are men. The main economic activity at Masese is fishing, followed by small businesses such as barber shops, boda-boda riding and casual labour. Like other fishing communities, the area also has a booming sex business and people paying more for condomless sex [24, 36].

The study was conducted among high-risk ABYM aged 10–24 years in Masese fishing community. ABYM were considered to be high-risk if they reported two or more concurrent sexual partners in the previous one year, or used alcohol or reported use of drugs before sex, with inconsistent or no condom use. Participants who were sick, drunk or high on drugs were excluded at the time of questionnaire administer. The sample size was calculated using the formula by Kish Leslie [37]. For this study, the (P) was estimated from risky sexual behavior among males in fishing communities in Uganda and it was 53.5% [38]. This was a proxy in a similar community of study as literature has shown that people who engaged in risky sexual behavior were more willing to use PrEP [11, 13]. The $Z\alpha/2$ (two-tailed normal distribution) was 1.96 (standard normal value at $\alpha = 5\%$ level of significance) and maximum error allowed was 5%.

Participants were selected by convenience sampling. This method was used because of the varying schedules of the participants making accessibility difficult where some fish at night and some during the day. The fishing communities are also characterized by high mobility and so other sampling methods would have given a high non-response rate. The village health team (VHT) and other ABYM helped in mobilization of the participants from their households and shops as well as from the residential areas, landing and mining sites. The VHT, interviewers and other ABYM moved to the different areas of the community informing all ABYM aged 10–24 years to convene at a given area. For the non-emancipated minors, the interviewers explained the study to the parents or guardians and sought their consent before engaging the boys. The participants were screened using the screening tool to determine if they were high-risk or not. In the screening tool, the potential participants were asked for the number of sexual partners, if they knew their partners' HIV status, used condoms during sexual intercourse and or used drugs or alcohol before sexual intercourse. Those who were high-risk constituted the sample.

## Measurement of variables

The dependent variable was willingness to take PrEP which was defined as a participant responding with 'likely' or 'very likely' when asked the question, "How likely is it that you would take PrEP to reduce your chances of getting HIV if it was provided?" This was a single question and the responses were on a 5-point likert scale- Very likely, likely, not sure, unlikely and very unlikely. Those who responded with 'very likely' and 'likely' were considered willing while those who responded with 'not sure', 'unlikely' and 'very unlikely' were considered not willing. The independent variables included socio-demographic characteristics (age, marital status, religion, highest level of education, occupation, and monthly income), HIV risk perception, and behavioral characteristics (number of sexual partners, condom use, drug and alcohol use, engagement in sex work, HIV testing and seeking for treatment when one had an STI). The participants gave a self-report of having an STI in the last twelve months, (both syndromic and laboratory confirmed). Risk-perception which was self-reported, was defined as the extent to which a participant considered themselves to be at risk of HIV infection and it was the main explanatory variable. The responses were on a 4-point Likert scale- Very high risk, At risk, not sure and Not at risk. Those who responded with very high risk and at risk were considered to perceive themselves at a risk of HIV acquisition while those who responded with not sure and not at risk were considered to perceive themselves not at risk of HIV acquisition. Alcohol and drug use were measured based on frequency of intake with very often being at least four times a week and sometimes once a week or once in a while. Data were also collected on PrEP attributes, which included likely potential barriers and facilitators among those who were deemed willing to take PrEP such as; mode of administration, safety in terms of side effects, accessibility in terms of location of PrEP services, and PrEP associated stigma. The responses were on a 5-point Likert scale- Very likely, likely, not sure, unlikely and very unlikely. Those who responded with very likely and likely were considered likely to take PrEP based on that attribute while those who responded with not sure, unlikely and very unlikely were considered unlikely to take PrEP.

## Data collection procedures and methods

Data were collected using semi-structured questionnaires through face to face interviews by trained research assistants who were bachelor's degree holders and fluent in both English and Lusoga, the local dialect in a place that allowed for privacy. Questionnaires for the study were developed in English and translated into Lusoga, the local language used at the landing site. The questionnaires were pre-tested with ABYM at Gaba, a fishing community around Kampala City to clarify any errors prior to using the tools. Data were collected on socio-demographic factors (age, marital status, religion, highest level of education, occupation, and monthly income), behavioral factors (Number of sexual partners, condom use, engagement in sex work, alcohol or drug use, sex under the influence of alcohol or drugs, HIV testing, STI treatment), willingness to take PrEP, and willingness to take PrEP in relation to the PrEP attributes informed by literature.

These were supervised by the principal investigator. The filled questionnaires were reviewed and checked for completeness, consistency, and legibility by the principal investigator during data collection, and feedback was provided to the research assistants every morning.

## Data handling and statistical analysis

Data were coded and entered into EpiData version 3.0 and exported to STATA version 16, for analysis (S1 Data). Two data entry clerks entered the data into epi data following double data

entry strategy. Data cleaning which included the checking of figures like age, structures and internal consistency was done during data entry. Continuous variables were summarized as either means (SD) if normally distributed or medians (IQR) if they were skewed. Categorical variables were summarized as frequencies and percentages. The outcome, "willingness to take PrEP" was expressed as a percentage. To determine the factors associated with willingness to take PrEP, bivariate analysis was carried out. Each independent variable was first analyzed by using logistic regression and all variables with a P-value <0.2 were considered for multivariable analysis. At multivariable analysis, modified Poisson regression was used and variables were dropped using the backward stepwise method of model building while controlling for confounding variables, leaving only variables that were significant at 5% cut off. Prevalence ratios and their 95% confidence intervals were used to express any association.

## Ethical clearance

Ethical approval was obtained from the Higher Degrees, Research and Ethics Committee, School of Public Health, Makerere University (IRB number FWA00011353). Permission was sought from the authorities of Jinja and Masese fishing community. Consent was sought from participants above 18 years of age and emancipated minors while assent was sought from non-emancipated minors as well as consent from their parent or guardians before administering the questionnaire. The research assistants went with the minors to their homes if they did not find them at the household to seek consent from the parents. Privacy and confidentiality were also observed during data collection by use of participant numbers and carrying out the interview away from other participants respectively. The participants were also informed that they could leave the study whenever they wanted and that participation was entirely voluntary. A brief description of PrEP was provided by word of mouth to potential participants by the interviewer prior to asking about willingness to take PrEP and its attributes.

## Results

### Participant socio-demographic characteristics

Table 1 shows the socio-demographic characteristics of survey Adolescent Boys and Young Men in Masese fishing community, Jinja district, (N = 479). The mean age of the participants was 20.9 years (SD = 2.5) and 77.4% were single. About one third of the respondents were Muslims (35.4%). More than half of the respondents had attained secondary level of education (56%) and 58.5% earned between USD 27 and USD 133. A total of 28.7% respondents engaged in fishing activities, 31.5% were casual laborers, while 20.3% engaged in small scale businesses.

### Behavioural characteristics of adolescent boys and young men in Masese fishing

The median number of sexual partners was 3 (IQR; 2–5) with 66.2% having 2–4 sexual partners and 33.8% having five or more sexual partners. More than half (59.3%) of the participants reported inconsistent condom use while 40.7% did not use condoms at all. Only half of the participants 50.3% had ever tested for HIV. With regard to the risk of acquiring HIV, 26.5% were not sure of their risk, 25.5% thought they were at risk, while 28% thought they were at very high risk (Table 2).

### Willingness to take PrEP and associated factors

The study found that 86.4% of the participants were willing to take PrEP. Perceived risk had a great bearing on willingness to take PrEP as willingness increased with increased perceived

**Table 1. Socio-demographic characteristics of survey adolescent boys and young men in Masese fishing community, Jinja district, (N = 479).**

| Variable | Frequency | Percentage (%) |
|---|---|---|
| **Mean age in years** (SD) | 20.9 (2.5) | |
| **Age group in years** | | |
| 10–19 | 153 | 31.9 |
| 20–24 | 326 | 68.1 |
| **Marital Status** | | |
| Single | 371 | 77.4 |
| Married | 102 | 21.3 |
| Separated | 6 | 1.3 |
| **Religion** | | |
| Catholic | 130 | 27.2 |
| Anglican | 79 | 16.5 |
| Muslim | 169 | 35.4 |
| Pentecostal | 90 | 18.8 |
| Others | 11 | 2.3 |
| **Education level** | | |
| None | 44 | 9.2 |
| Primary | 151 | 31.5 |
| Secondary | 268 | 56 |
| Tertiary | 16 | 3.3 |
| **Monthly income (USD)** | | |
| <27 | 175 | 36.5 |
| 27.1–133 | 280 | 58.5 |
| >133 | 24 | 5 |
| **Occupation** | | |
| Fishing activities | 137 | 28.7 |
| Businessmen | 97 | 20.3 |
| Casual labourers | 150 | 31.5 |
| Unemployed | 24 | 4.5 |
| Others | 71 | 15 |

risk. Among those who thought they were at very high risk of HIV acquisition, 90.3% were willing to take PrEP while among those who self-perceived to be at no risk for HIV acquisition, the willingness to take PrEP was 76%. Of the 414 participants, 84.3% would be willing to take PrEP even if they had to pay for it, 95.4% if offered in oral form; 96.6% if there was information on PrEP and 87.7% if there was evidence that PrEP works as well as if PrEP was available in areas they could easily access it.

## Bivariate and multivariable analysis of factors associated with willingness to take PrEP

After adjusting for potential and suspected confounders, the regression analysis showed that the factors associated with willingness to take PREP were very high-risk perception, availability of PrEP in an accessible place, confidence in PrEP safety, monthly income and marital status.

Table 3 also shows the socio demographic and behavioral factors associated with willingness to take PrEP. Those who perceived themselves to be at very high risk of HIV acquisition were 1.11 times (adjusted prevalence ratio [adj. PR] = 1.11; 95% Confidence Interval [95%CI]:

**Table 2. Behavioural characteristics of adolescent boys and young men in Masese fishing community, Jinja district, (N = 479).**

| Variable | Frequency | Percentage |
|---|---|---|
| **Condom use in the last 12 months** | | |
| Not at all | 195 | 40.7 |
| Sometimes | 284 | 59.3 |
| **Drug use in the last 12 months** | | |
| Very often | 39 | 8.1 |
| Sometimes | 55 | 11.5 |
| Never | 385 | 80.4 |
| **Alcohol use in the last 12 months** | | |
| Very often | 61 | 12.7 |
| Sometimes | 125 | 26.1 |
| Never | 293 | 61.2 |
| **Sex under drug influence** | | |
| Yes | 65 | 13.6 |
| No | 414 | 86.4 |
| **Sex under alcohol influence** | | |
| Yes | 124 | 25.9 |
| No | 355 | 74.1 |
| **Had an STI in the past 12 months** | | |
| Yes | 163 | 34 |
| No | 316 | 66 |
| **Went to hospital for STI\*** | | |
| Yes | 135 | 82.8 |
| No | 28 | 17.2 |
| **Did an HIV test in the past 12 months** | | |
| Yes | 238 | 49.7 |
| No | 241 | 50.3 |
| **HIV risk perception** | | |
| Very high risk | 134 | 28 |
| At risk | 122 | 25.5 |
| Not sure | 127 | 26.5 |
| Not at risk | 96 | 20 |

\*Expressed out of those who reported a STI

1.03, 1.20) more likely to be willing to take PrEP compared to those who thought they were not at risk of acquiring HIV. However, those who earned more than USD 27 were 92% (adj. PR = 0.92; 95% CI: 0.87, 0.97) less likely to be willing to take PrEP compared to those who earned less than USD 27 and ABYM who were single were 92% (adj. PR = 0.92; 95% CI: 0.87, 0.98) less likely to be willing to take PrEP compared to those who were married.

Table 4 shows the PrEP attributes associated with willingness to take PrEP. Compared to those who were unlikely to take PrEP despite the confidence in its safety, ABYM who were likely to take PrEP if they were confident in PrEP safety were 1.40 times (adj. PR = 1.40; 95% CI: 1.25, 1.57) more willing to take PrEP. Also, ABYM who were likely to take PrEP if it were available in an area they could easily access it were 1.86 times (adj. PR = 1.86; 95% CI: 1.55, 2.24) more likely be willing to take PrEP compared to those who were unlikely to take PrEP despite its availability in an area they can easily access it.

**Table 3. The socio demographic and behavioural factors associated with willingness to take PrEP among ABYM in Masese fishing community (N = 479).**

| Variable | N = 479 | Willingness to take PrEP n(%) | | cPR (95%CI) | Adj.PR (95%CI) |
|---|---|---|---|---|---|
| | | Willing | Not willing | | |
| | **Risk perception of HIV acquisition** | | | | |
| Not at risk | 96 | 73(76.0) | 23 (24) | 1 | 1 |
| Not sure | 127 | 111(87.4) | 16 (12.6) | 1.15(1.01,1.31) | 1.09(0.99,1.18) |
| At risk | 122 | 109(89.3) | 13 (10.7) | 1.17(1.03,1.34) | 1.05(0.97,1.14) |
| Very high risk | 134 | 121(90.30) | 13 (9.7) | 1.19(1.05,1.35) | **1.11(1.03,1.20)** |
| | **Age group (years)** | | | | |
| 10–19 | 153 | 124(81.0) | 29 (19) | 1 | 1 |
| 20–24 | 326 | 288(88.3) | 38 (11.7) | 1.07(0.98,1.17) | 1.05(0.99,1.11) |
| | **Marital status** | | | | |
| Married | 102 | 93(91.2) | 9 (8.8) | 1 | 1 |
| Single | 377 | 321(85.1) | 56 (14.9) | 0.93(0.86,1.01) | **0.92(0.87,0.98)** |
| | **Religion** | | | | |
| Anglican | 79 | 67(84.8) | 12 (15.2) | 1 | |
| Catholic | 130 | 118(90.8) | 12 (9.2) | 1.07(0.96,1.19) | |
| Muslim | 169 | 146(86.4) | 23 (13.6) | 1.02(0.91,1.14) | |
| Others | 101 | 83(82.2) | 18 (17.8) | 0.97(0.85,1.10) | |
| | **Education level** | | | | |
| None | 44 | 32(72.7) | 12 (27.3) | 1 | 1 |
| Primary | 151 | 131(86.8) | 20 (13.2) | 1.19(0.98,1.44) | 1.05(0.91,1.20) |
| Secondary and above | 284 | 251(88.4) | 33 (11.6) | 1.22(1.01,1.46) | 1.05(0.91,1.20) |
| | **Monthly income** | | | | |
| <USD 27 | 175 | 157(89.7) | 18 (10.3) | 1 | 1 |
| USD27 + | 304 | 257(84.5) | 47 (15.5) | 0.94(0.88,1.01) | **0.92(0.87,0.97)** |
| | **Number of sexual partners in the last 12 months** | | | | |
| 5+ | 162 | 139(85.8) | 23 (14.2) | 1 | |
| 2–4 | 317 | 275(86.8) | 42 (13.2) | 1.01(0.94,1.09) | |
| | **Condom used in the last 12 months** | | | | |
| Not at all | 195 | 167(85.6) | 28 (14.4) | 1 | |
| Sometimes | 284 | 247(87.0) | 37 (13.0) | 1.02(0.94,1.09) | |
| | **Had an STI in the last 12 months** | | | | |
| Yes | 163 | 148(90.8) | 15 (9.2) | 1 | |
| No | 316 | 266(84.2) | 50 (15.8) | 0.93(0.870.99) | |
| | **Did an HIV test in the last 12 months** | | | | |
| Yes | 238 | 213(89.5) | 25 (10.5) | 1 | |
| No | 241 | 201(83.4) | 40 (16.6) | 0.93(0.87,1.00) | |
| | **Occupation** | | | | |
| Unemployed | 24 | 22(91.7) | 2 (8.3) | 1 | |
| Businessmen | 97 | 82(84.5) | 15 (15.5) | 0.93(0.79,1.09) | |
| Fishing activities | 137 | 122(89.1) | 15 (10.9) | 0.98(0.85,1.13) | |
| Casual labourers | 150 | 127(84.7) | 23 (15.3) | 0.93(0.80,1.08) | |
| Others | 71 | 61(85.9) | 10 (14.1) | 0.95(0.80,1.11) | |

**Table 4. Association between PrEP attributes and willingness to take PrEP among ABYM in Masese fishing community (N = 414).**

| Variable | N = 414 | Willingness to take PrEP n(%) | cPR (95%CI) | aPR (95%CI) |
|---|---|---|---|---|
| Likelihood take PrEP even if they had to pay for it* | | | | |
| Unlikely | 65 | 65(15.7) | 1 | |
| Likely | 349 | 349(84.3) | 2.00(1.68,2.37) | |
| Likelihood to take PrEP if it was available in injectable form* | | | | |
| Unlikely | 108 | 108(26.1) | 1 | |
| Likely | 306 | 306(73.9) | 1.60(1.43,1.80) | |
| Likelihood to take PrEP if it was available in oral form* | | | | |
| Unlikely | 19 | 19(4.6) | 1 | |
| Likely | 395 | 395(95.4) | 4.42(2.98,6.57) | |
| Likelihood to take PrEP if information on PrEP was available* | | | | |
| Unlikely | 14 | 14(3.4) | 1 | |
| Likely | 400 | 400(96.6) | 5.64(3.51,9.08) | |
| Likelihood to take PrEP if you were confident that it was effective* | | | | |
| Unlikely | 51 | 51(12.3) | 1 | |
| Likely | 363 | 363(87.7) | 2.27(1.85,2.79) | |
| Likelihood to take PrEP if they were sure that it was safe* | | | | |
| Unlikely | 74 | 74(17.9) | 1 | **1** |
| Likely | 340 | 340(82.1) | 1.88(1.61,2.20) | **1.40(1.25,1.57)** |
| Likelihood to take PrEP if it was available at a place they could easily get it* | | | | |
| Unlikely | 51 | 51(12.3) | 1 | **1** |
| Likely | 363 | 363(87.7) | 2.27(1.85,2.79) | **1.86(1.55,2.24)** |
| Likelihood to take PrEP if they were sure no one would call them HIV positive* | | | | |
| Unlikely | 95 | 95(22.9) | 1 | |
| Likely | 319 | 319(77.1) | 1.68(1.48,1.91) | |

## Discussion

This study sought to determine willingness to take PrEP and associated factors among high-risk adolescent boys and young men in Masese fishing community, Uganda.

This study found that 86.4% of the participants were willing to take PrEP. This willingness is higher than the levels of willingness found by other researches among key and priority populations; 46.5% by Lee, Chang et al., (2017), 55% by Holloway et al., (2017) among MSM in California, and 57.6% found by Ferrer et al., (2016) among MSM in Spain [9, 11, 12]. Our result is however close to that found by Peng, Cao et al., (2019) at 84.9% [13] and by Ssuna et al., in a fishing community in Uganda at 80.6% [14] but lower than the one found among adolescents (13–18 years) where it was 90.4% [10]. This variation could be due to inherent differences in attitudes and knowledge of the populations studied and their settings with majority being resource rich settings. Nonetheless, the high willingness among adolescent boys and young men therefore means PrEP may be able to greatly contribute towards HIV prevention efforts to reduce HIV incidence in this population.

The study found that when people perceived themselves to be at very high risk of HIV acquisition, they would be more willing to take PrEP compared to those who did not perceive themselves to be at risk of HIV acquisition. This is because they thought they were more likely to acquire HIV compared to those who thought they were not at risk. This agrees with findings from other studies that found that individuals who perceived themselves to be at risk for HIV were more likely to take PrEP even among adolescents [9, 14, 22, 39–41]. Holloway et al.,

(2017) found that those who rated their risk of HIV contraction as medium or high would be more willing to take PrEP [11]. Young et al., (2013) found that those who deemed themselves not at risk were less likely to take PrEP [42]. Peng et al., (2019) attributed this risk estimation to engaging in risky sexual behavior [13] which was one of the criteria for being included in the study. This therefore means that willingness to take PrEP is high among high-risk groups and these should be identified and targeted for prevention using PrEP.

Confidence that PrEP was safe also had a positive bearing on willingness to take PrEP. The findings showed that people were more willing to take PrEP if they were confident that PrEP was safe. The findings agree with a study by Ferrer et al., (2016) who found that men disagreed or strongly disagreed that they would use PrEP if there were side effects in general [9]. Similar findings were got in Uganda where the participants who included fisher folk and were adolescents and young people [20, 22]. This means that the men would be more willing to take PrEP if they were confident in its safety in terms of side effects [9]. A mini review also showed that PrEP safety was found to be associated with willingness to take PrEP [17]. This could be due to the fact that people would want to continue with their day to day life as side effects such as vomiting, nausea or dizziness would affect their productivity hence posing as a potential barrier to willingness. PrEP service providers should therefore speak openly and honestly to ABYM considering PrEP about PrEP side effects and what to do if side effects occur.

Access to PrEP was also found to be associated with willingness to take PrEP. The availability of PrEP in areas that could be easily accessed by ABYM increased willingness to take PrEP. These findings agree with a study that was conducted in the rural areas of the Midwest where it was noted that the primary impediment to PrEP adoption was a lack of PrEP infrastructure in rural areas. In the Midwest, participants described their localities as PrEP deserts since there were no PrEP clinics in their immediate or adjacent areas [43]. This setting is similar to most of the fishing communities which are in rural areas and cannot easily access health services [44]. Also, Muhumuza et al., whose study was on young people, Muwonge et al., and Ddaaki et al., whose studies had fisherfolk also found that access to PrEP was associated with willingness to take PrEP. These people are busy and so moving long distances which is time consuming and tiring to the PrEP users may make them unwilling to take PrEP [20–22]. Therefore, programs should aim at making PrEP accessible in fishing communities as access has been shown to be a predictor of willingness to take PrEP.

Monthly income was found to be negatively associated with willingness to take PrEP. Participants who earned more than USD 27 were less willing to take PrEP compared to those who earned less than USD 27. These findings contradict a study by Zhang et al., (2013) which showed that participants with USD 149–446 were more willing to take PrEP compared to those of lower income levels [41]. This could have been because most of those who earned more than USD 27 deemed themselves not at risk of acquiring HIV compared to those who earned less than USD 27. About 21.1% of those who earned more than USD 27 deemed themselves not at risk compared to 18.3% of those who earned less than USD 27 who deemed themselves not at risk. This could also be due to the fact that people with more income can afford other HIV prevention measures such as buying condoms. These are however probable explanations and therefore calls for more research to determine if income has an association with willingness to take PrEP because income is a cue to action as people with higher income would ideally be more willing to take PrEP since they can afford to buy even if it came at a price.

We also found that marital status was negatively associated with willingness to take PrEP. Those who were unmarried were less willing to take PrEP compared to those who were married. These findings are similar to those found by Zhang et al., (2013) where willingness was found to be more among those who were married [41]. This could have been because most of those who were unmarried deemed themselves not at risk of contracting HIV compared to

those who were married. This could be because those who were single did not feel answerable to anyone and would therefore not feel accountable. Another possible explanation could be drawn from the study findings. According to the findings in this study, 20.4% of those who were unmarried deemed themselves not at risk of HIV acquisition while 18.6% of those who were married deemed themselves not at risk of HIV acquisition. This therefore means that a higher number of those who were single thought they were less likely to contract HIV and therefore the willingness to take PrEP would be lower. This too is a probable explanation as most of the available studies do not directly talk about marital status. There is therefore need to carry out further studies to determine if marital status has a bearing on willingness to take PrEP.

## Study limitations and strengths

Social desirability bias may also have influenced how participants responded to questions, given the sensitive nature of some topics such as monthly income, number of sexual partners, condom use resulting in alteration of willingness to take PrEP. To address this, we ensured the interviews were carried out in private so that the participants could open up and give unbiased information. Using convenience sampling method could have introduced selection bias. This was mitigated by ensuring that all the zones on the island had representation and sample size large enough. Since the study was looking at a twelve months' period, there is a possibility of recall bias. The interviewers were patient with the participants as they recalled in response to the questions and focused on more recent events such as condom use with the most current partner before probing the participants to recall 12 months.

## Conclusion

This study has demonstrated high willingness (86.4%) to take PrEP among adolescent boys and young men in Masese fishing community. ABYM who perceived themselves to be at high risk for HIV acquisition, could easily access PrEP and were confident in PrEP safety were significantly more likely to be willing to take PrEP while those who earned more than USD 27 and those unmarried were significantly less likely to be willing to take PrEP. More research however is also needed to establish the relationship between monthly income and marital status with willingness to take PrEP.

## Supporting information

**S1 Data.**
(DTA)

## Acknowledgments

We extend our gratitude to the district health office of Jinja as well as the leaders and adolescent boys and young men of Masese fishing community for the permission to conduct the study and all the support in this study respectively. We would also like to extend our sincere thanks to the research assistants; Akol Catherine, Mutegule Shafiq and Tigaiza Arnold who worked tirelessly to see to it that quality data is collected on time. We also would like to thank the school of public health for the supervision and the Makerere university- Behavioural and Social Science Research program at the college of health sciences for the mentorship.

## Author Contributions

**Conceptualization:** Winnie Agwang, Fred C. Semitala, Simon Kasasa, Joseph K. B. Matovu.

**Data curation:** Winnie Agwang.

**Formal analysis:** Winnie Agwang, Joseph K. B. Matovu.

**Funding acquisition:** Winnie Agwang, Fred C. Semitala, Joseph K. B. Matovu.

**Investigation:** Winnie Agwang, Sherifah Nabikande, Tom Okello, Joan Tusabe, Joseph K. B. Matovu.

**Methodology:** Winnie Agwang, Simon Kasasa, Joseph K. B. Matovu.

**Project administration:** Joanita Nangendo.

**Resources:** Joanita Nangendo, Fred C. Semitala.

**Supervision:** Winnie Agwang, Joanita Nangendo, Simon Kasasa, Joseph K. B. Matovu.

**Validation:** Winnie Agwang, Joanita Nangendo, Sherifah Nabikande, Tom Okello, Joan Tusabe, Simon Kasasa, Joseph K. B. Matovu.

**Visualization:** Tom Okello, Joseph K. B. Matovu.

**Writing – original draft:** Winnie Agwang, Joanita Nangendo, Sherifah Nabikande, Tom Okello, Joan Tusabe, Fred C. Semitala, Simon Kasasa, Joseph K. B. Matovu.

**Writing – review & editing:** Winnie Agwang, Joanita Nangendo, Sherifah Nabikande, Tom Okello, Joan Tusabe, Fred C. Semitala, Simon Kasasa, Joseph K. B. Matovu.

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
