## [Decision Letter · Decision Letter 0]

2 Jan 2023

PGPH-D-22-01436

Factors associated with willingness to take Pre-Exposure Prophylaxis (PrEP) among high-risk adolescent boys and young men in Masese Fishing Community, Uganda

Dear Dr. Agwang,

Thank you for submitting your manuscript to PLOS Global Public Health. After careful consideration, we feel that it has merit but does not fully meet PLOS Global Public Health’s publication criteria as it currently stands. Therefore, we invite you to submit a revised version of the manuscript that addresses the points raised during the review process.

We look forward to receiving your revised manuscript.

Kind regards,

Henry Zakumumpa, PhD

Academic Editor

Journal Requirements:

1. Please amend your Data Availability Statement and indicate where the data may be found.

Additional Editor Comments (if provided):

We are delighted to share reports from our reviewers. Please make a point-by-point response to each of the comments raised as much as possible so can move swiftly to a decision.

I would like to make special mention of the need to especially address comments around methodology such as the points raised around sampling.

Also, endevour to use as much local evidence from Uganda as raised by both reviewers. Lastly, there have been several comments raised around the discussion which needs a revision in line with the reviewers' suggestions.

Reviewers' comments:

Reviewer's Responses to Questions

**Comments to the Author**

1. Does this manuscript meet PLOS Global Public Health’s publication criteria? Is the manuscript technically sound, and do the data support the conclusions? The manuscript must describe methodologically and ethically rigorous research with conclusions that are appropriately drawn based on the data presented.

Reviewer #1: Yes

Reviewer #2: Partly

2. Has the statistical analysis been performed appropriately and rigorously?

Reviewer #1: Yes

Reviewer #2: Yes

3. Have the authors made all data underlying the findings in their manuscript fully available (please refer to the Data Availability Statement at the start of the manuscript PDF file)?

Reviewer #1: Yes

Reviewer #2: Yes

4. Is the manuscript presented in an intelligible fashion and written in standard English?

Reviewer #1: Yes

Reviewer #2: Yes

5. Review Comments to the Author

Reviewer #1: Major comments

Introduction:

In the first paragraph authors mention that PrEP has been effective in prevention of HIV in high-risk populations, it would be helpful to the readers to quantify how effective it is by citing studies which have been done around Uganda or East Africa if any.

In the second paragraph, the authors report willingness to use PrEP in high-risk populations and MSM in China. It would be important to cite this willingness among the studied population (fishing community) and also cite studies near Uganda to bring similarities in location, culture and HIV risk.

In the last paragraph, the authors mention that understanding willingness to take PrEP among adolescent boys and young men in a fishing community will provide essential pre-intervention information that can guide the design and implementation of PrEP interventions for ABYM in other fishing communities. Which is a good point, however, this needs more clarity. The readers would benefit to understand why the authors chose ABYM in the entire population, what is special about them compared to the general population?

Methods

In the study setting (first paragraph), the authors did a good job of mentioning the population of the area and prevalence of HIV. Is this prevalence for Masese or fishing communities in general? It would be important to the readers to know why the authors went for Masese and what proportion of the population are ABYM?

In the second paragraph the authors describe how they calculated the sample size of the study. However, they did not provide the readers with the power calculations of the Poisson regression which was used for associated factors.

In the third paragraph, the authors describe convenience sampling as a method of sampling participants used. Why did the authors decide to use a non-probability sampling method in such a community over other methods like systematic sampling? Wouldn’t this lead to selection bias for only those who were willing to convene in the suggested place?

In the section of measurement of variables, the authors said the primary outcome was willingness to use pre-exposure prophylaxis (PrEP) which was defined as a participant responding with ‘likely’ or ‘very likely’. However, this did not create a clear-cut difference in those who are likely to use PrEP and whose who were unlikely to use PrEP since both answers are positive. The readers would benefit from explaining to them how the negative responses of those who don’t want PrEP or unlikely to use PrEP was achieved.

Results

In Table 2, how did the authors differentiate between “very high risk” and “at risk” while asking the participants?

In Table 3 and 4, the authors reported a single proportion of willingness to use PrEP. The readers would benefit to know the two sides of willingness to use PrEP since it was measured as “likely” and “very likely”? Who were willing and unwilling to use PrEP?

In Table 4, how did the authors differentiate between “likely” and “unlikely” responses and the outcome of willingness to use PrEP which was measured as “likely” and “very likely”?

Limitations

The study did not intend to find out causal effect, therefore it cannot be a limitation as mentioned in the first sentence

One limitation would be the gender role which may play an influence of men to use PrEP which was no captured in this study

Minor comments

Introduction

In the last sentence of paragraph one the authors introduce a concept of adherence are reported that it is directly proportional to willingness to take PrEP. The study which was cited is a randomized trial which doesn’t bring out the picture

In the last sentence of paragraph two, the authors mention about increase in men’s risk of HIV transmission. This sentence needs more specificity, the readers would benefit from more information for example what is the transmission rate compared to Uganda general population?

Methods

In the section of ethics, the authors would explain more on how they assented the minors. From the sampling technique used, did the minors come with their guardians at the place of convince? And how was confidentiality achieved?

Conclusions and recommendations

In the last sentence, the authors mention this shows that adolescent boys and young men in fishing communities are a suitable population for HIV prevention. However, this study was not aimed at determining suitability of the population. Readers would benefit in conclusions and recommendations around the study aim.

Reviewer #2: General comments; The manuscript addresses an important topic among high risk young men who are often times not the focus of research yet interventions such as oral PrEP are beneficial to this group and their sexual partners.

Introduction:

1. Lines 71-72 give some background data on willingness among men outside Africa. Do also compare with studied in Africa or other young people e.g., adolescent girls and young women (AGYW). This study doi:10.1002/jia2.25909 (done among AGYW in Kampala has results on willingness to take oral PrEP even though its focus was mainly uptake and adherence.

2. Line 78 mentions different populations, yet studies referenced in lines 73-78 are mainly among MSM with one done among opium users. Qualify this statement by highlighting which willingness factors are unique to MSM, drug users, fisherfolk etc and which factors are commonly found in most high-risk male populations (would also be the same factors for females). You can also read the literature to find if there are differences between younger men like ABYM and older men.

3. Lines 83-85, fisherfolk also have disposable income so usually go for sex workers since they can afford to pay.

4. Lines 86-88; Keep consistency in the decimal place for results written in percentages.

5. The gap is not well articulated at the end of the introduction. Before the authors mention “understanding willingness”, they should mention what the gap in literature is that is driving the research question. This is related to point 1 above, if the authors have searched the literature on willingness among ABYM and it is lacking, do mention it.

Methods:

6. Line 110, The title could include sampling methods rather than sample size. Details of the sample size calculation are better suited for the study protocol but sampling methods used in the study are important to include here.

7. Lines 118-119; This statement does not describe the study setting but rather a general statement. “However, PrEP in fishing communities is 119 not yet fully embraced despite having HIV prevalence as high as 37%” It could be why you want to study willingness so can be taken to the introduction.

8. The sample of ABYM include minors, how did the VHTs and other ABYM reach these boys without their parents/ guardians involved. Briefly clarify what the process was for those who were <18 years. This is mentioned in ethics section but also needs a brief here on emancipated minors, parental consent etc.

9. Line 138, qualify the partner’s status by specifying “HIV”

10. Lines 158-162 repeat information already given in the section before on study variables and, also given below in the “measurement of variables”. Re-do or merge some of these sections to avoid repetition.

11. Line 169; give the whole range of possible responses on the measurement scale to give readers a clue of what the responses were for participants classified as “unwilling”

12. Line 171, what was the cut-off or categories to define perceived risk and no-perceived risk? Please mention this for clarity so that it is understood when the reader goes to the results section.

13. Study variables: mention whether the STI diagnosis was syndromic approach or laboratory tests were done. If labs were done, describe the assay under study procedures. Ensure that it is clear how you measured all the other variables.

14. Line 178; was there any double entry and data cleaning? Single entry in EpiData may have errors that go unnoticed if no active cleaning is done.

15. Line 179, add “if normally distributed” after means otherwise it looks like “if they were skewed” applies to both the means and medians.

16. Lines 187-188: Check the last sentence, it doesn’t read well/ is incomplete.

17. Lines 190-198; Did this study not require approval by the national council (UNCST)?

Results

18. Lines 224-225; Rephrase this sentence and include the chi square p-value. The results show that a higher proportion of those who perceived themselves at risk of HIV were willing to take PrEP compared to those who had lower risk perception. The results are proportions and don’t show that willingness increased with higher risk perception.

Discussion

19. Line 263; Rather than repeat results, mention if this level of willingness is high/ low/ expected compared to other literature

20. Line 268 is a repetition of lines 263-264. Delete one.

21. Lines 268-271; The willingness in your study is much higher than studies you qualify as having results in the same range. Revisit this statement and interpret results correctly in relation to other studies.

22. Lines 272-273; The results of these 2 studies and yours are in the same range compared to those in lines 268-271. Improve the discussion here. Discuss these studies with high willingness well in relation to yours (especially reference 36). Looks like young boys/ adolescents have high willingness.

23. Line 326; Those earning more money are also more likely to afford condoms and therefore rightly perceived themselves at lower risk which supports their unwillingness since they already use condoms for HIV prevention. Enrich your discussion further using literature.

24. Lines 3545-346; Look within your data and see if those who had high risk perception and reported high risk behaviors also had STI symptoms. Although STIs are sometimes asymptomatic, corroborating these variables within your data may give a clue to inconsistencies and if desirability bias was really a major issue or it occurred at a lower frequency.

Referencing

Use Vancouver style as in the author guidelines and read instructions to authors to ensure all manuscript sections conform to journal requirements.

6. PLOS authors have the option to publish the peer review history of their article (what does this mean?). If published, this will include your full peer review and any attached files.

**Do you want your identity to be public for this peer review?** For information about this choice, including consent withdrawal, please see our Privacy Policy.

Reviewer #1: **Yes: **BASHIR SSUNA

Reviewer #2: No

---

## [Decision Letter · Decision Letter 1]

16 May 2023

Factors associated with willingness to take Pre-Exposure Prophylaxis (PrEP) among high-risk adolescent boys and young men in Masese Fishing Community, Uganda

PGPH-D-22-01436R1

Dear Winnie Agwang,

We are pleased to inform you that your manuscript 'Factors associated with willingness to take Pre-Exposure Prophylaxis (PrEP) among high-risk adolescent boys and young men in Masese Fishing Community, Uganda' has been provisionally accepted for publication in PLOS Global Public Health.

Best regards,

Henry Zakumumpa, PhD

Academic Editor

We wish to thank the authors for attending to the comments of our reviewers. We are pleased to editorially accept this paper.

Reviewer Comments (if any, and for reference):

Reviewer's Responses to Questions

**Comments to the Author**

1. If the authors have adequately addressed your comments raised in a previous round of review and you feel that this manuscript is now acceptable for publication, you may indicate that here to bypass the “Comments to the Author” section, enter your conflict of interest statement in the “Confidential to Editor” section, and submit your "Accept" recommendation.

Reviewer #1: All comments have been addressed

Reviewer #2: All comments have been addressed

2. Does this manuscript meet PLOS Global Public Health’s publication criteria? Is the manuscript technically sound, and do the data support the conclusions? The manuscript must describe methodologically and ethically rigorous research with conclusions that are appropriately drawn based on the data presented.

Reviewer #1: Yes

Reviewer #2: Yes

3. Has the statistical analysis been performed appropriately and rigorously?

Reviewer #1: Yes

Reviewer #2: Yes

4. Have the authors made all data underlying the findings in their manuscript fully available (please refer to the Data Availability Statement at the start of the manuscript PDF file)?

Reviewer #1: Yes

Reviewer #2: Yes

5. Is the manuscript presented in an intelligible fashion and written in standard English?

Reviewer #1: Yes

Reviewer #2: Yes

6. Review Comments to the Author

Reviewer #1: The authors have responded to most of the comments and concerns.

However, in Table 4 is a confusing to the readers, the authors introduced a new term of "likelihood" which is ambiguous in statistical terms. It would have been better if the authors stick to the outcome wording of "willingness". For example "Willingness to take PrEP if they had to pay for it" and outcome can remain "Willing or Unwilling"

Reviewer #2: (No Response)

7. PLOS authors have the option to publish the peer review history of their article (what does this mean?). If published, this will include your full peer review and any attached files.

**Do you want your identity to be public for this peer review?** For information about this choice, including consent withdrawal, please see our Privacy Policy.

Reviewer #1: **Yes: **BASHIR SSUNA

Reviewer #2: No
